# Comparison of Patient-Reported Outcomes Measures and Quality-Adjusted Life Years Following One- and Two-Stage Septic Knee Exchange

**DOI:** 10.3390/antibiotics11111602

**Published:** 2022-11-11

**Authors:** Maximilian Budin, Salahulddin Abuljadail, Giacomo Traverso, Seper Ekhtiari, Thorsten Gehrke, Rachel Sommer, Mustafa Citak

**Affiliations:** 1Helios ENDO-Klinik Hamburg, 22767 Hamburg, Germany; 2Institute for Health Services Research in Dermatology and Nursing (IVDP), University Medical Center Hamburg-Eppendorf (UKE), 20246 Hamburg, Germany

**Keywords:** patient-reported outcome measures, periprosthetic joint infection, one-stage, two-stage, total knee arthroplasty, quality adjusted life years

## Abstract

(1) Background: Periprosthetic joint infection (PJI) can be managed with one- or two-stage revision surgery protocol. Despite several studies analyzing the eradication rates between both procedures, there are no comparative studies that analyze patient-reported outcome measures (PROMs) and quality-adjusted life years (QALYs) in both treatment strategies. (2) Methods: All patients who underwent a two-stage knee revision between January 2017 to December 2018, due to a periprosthetic joint infection were included in the study. From the time interval, we selected a comparative group with the one-stage septic procedure. All patients received the following questionnaires: Oxford Knee Score, EQ-5D-5L, SSQ-8, and the SF-36. Additionally, demographic patient data were collected. The quality-adjusted life years (QALY) were calculated using the EQ-5D-5L. (3) Results: A total of 35 patients with a mean age of 67.7 years (SD = 8.9) were included in the final evaluation. The mean follow-up period was 54.5 months (SD = 5.5). There was no statistically significant difference regarding the Charlson Comorbidity Index (CCI), postoperative complications, or all evaluated questionnaires. There was no statistically significant difference in QALYs between the one- and two-stage revision. (4) Conclusion: Our study results show that the one-stage revision for PJI achieves similar PROMs compared to two-stage revision.

## 1. Introduction

Periprosthetic joint infection (PJI) following total joint arthroplasty (TJA) is a serious complication that almost always requires revision surgery [1]. In chronic PJI, exchange arthroplasty is the method of choice. The exchange arthroplasty can be performed as a one- or two-stage procedure. Two-stage exchange arthroplasty involves at minimum two separate operations, with the first involving removal of all implants, thorough irrigation and debridement, and implantation of a temporary ‘spacer’ implant. The patient is then placed on intravenous antibiotics, and followed clinically for evidence of infection resolution, at which time the second stage is planned for implantation of a definitive implant.

Similar to the two-stage procedure, the therapeutic target of the one-stage technique is the eradication of the infection, but with only one surgical intervention in carefully selected patients. According to current literature, medium- to long-term success, following the one-stage revision ranges from 75% to 98% [2,3,4,5,6,7,8,9]. A recently published systematic review by Pangaud et al. showed that the one-stage method is a viable alternative to the two-stage revision with similar eradication rates [10], while the one-stage procedure was associated with lower cost and morbidity compared to the two-stage method [10].

Despite several studies focusing on clinical outcomes, there are no comparative studies that report on the patient-reported outcome measure (PROMs) and quality-adjusted life years (QALYs) in septic revision knee procedures. Focusing on hip procedures, Tirumala et al. (2021) were able to show significantly better results for PROMs in patients with a one-stage septic hip revision compared to the two-stage revision [11].

The aim of this current study was to find out if PROMs differ between the one-stage and two-stage revision for septic TKAs. We hypothesized that the one-stage septic knee exchange might be associated with a higher PROMs and QALYs compared to the two-stage septic knee exchange.

## 2. Material and Methods

At our institution, we mainly perform one-stage septic exchange arthroplasty in the management of PJI. Therefore, all patients who were surgically treated with the two-stage septic knee revision in the period from 1 January 2017 to 31 December 2018 were initially selected (47 patients). The exclusion criteria were: patients treated surgically at another hospital, patients without complete documentation, patients without preoperative aspiration to determine the microorganism, and patients with a follow-up period of less than 3 years. At our institution, a PJI is diagnosed using the 2018 PJI criteria by Parvizi et al. [12].

A total of 36 patients met the inclusion and exclusion criteria. During the same period, 47 patients who underwent a one-stage septic knee revision were randomly selected. The same inclusion and exclusion criteria were set. Patients were matched for age and pre-existing conditions.

The following data were collected for all patients: age, gender, weight (in kg), height (in cm), body mass index (BMI), hypertension, diabetes mellitus, admission diagnosis, Charlson Comorbidity Index (CCI), hospital stay, number of germs, type of germs (high virulent, low virulent), postoperative complications/re-revisions during the inpatient stay and at the time of the latest follow up, and postoperative mortality during the inpatient stay and at the time of the latest follow up.

In addition, a correlation analysis between one-stage and two-stage revision was performed. The correlation between the two procedures was examined for the following variables: gender, number of germs, high virulent vs. low virulent germs, gram positive vs. gram negative germs, postoperative deep vein thrombosis (DVT), postoperative pulmonary embolism, and admission to the intensive care unit (ICU).

The patients’ function and health-related quality of life for both procedures was determined by PROMs. For this purpose, the following standardized and well-studied questionnaires were sent to all patients by post: Oxford Knee Score (OKS), European Quality of Life 5 Dimensions 5 Level Version (EQ-5D-5L), Surgical Satisfaction Questionnaire-8 (SSQ-8), and Short From 36 (SF-36). To determine the quality-adjusted life years (QALY), the EQ-5D-5L index was used. The QALYs are calculated as followed: QALYs = Follow-up (in years) × EQ-5D-5L Index.

A total of 35 patients responded (response rate of 53%). Patients who did not respond to the letter were contacted by telephone. At the time of the last survey, a total of 7 patients had died (4 patients in the one-stage group and 3 patients in the two-stage group). No information on the causes of death were provided by the relatives. The remaining patients could not be reached due to a change of telephone number and/or address. Therefore, the final cohort included 35 patients (*n* = 19 one-stage, *n* = 16 two-stage) with a minimum follow-up of three years.

This study was performed after obtaining approval from the local ethics committee.

### 2.1. Surgical Technique

#### 2.1.1. One-Stage Exchange Arthroplasty Technique

The one-stage exchange technique to treat PJI was first described by Buchholz in 1970s, after mixing antibiotics with bone cement to enhance the local antibiotics concentrations [13]. The success of this technique depends on the complete removal of all implanted hardware, cement removal, and radical debridement with excision of infected and necrotic soft and bone tissue. Re-draping the patient and re-scrubbing the surgical team must be performed with the use of new instruments for re-implantation. The indication for one-stage septic exchange technique depends on the preoperative diagnostic joint aspiration after recognition of the causative microorganism and obtaining the antibiotic sensitivity results. Routinely, postoperative 2 week-intravenous antibiotic therapy is administered [14].

#### 2.1.2. Two-Stage Exchange Arthroplasty Technique

The two-stage exchange technique is considered the gold standard worldwide for treating chronic PJI. It was first described in 1983 by Insall et al. [15]. A preoperative diagnostic joint aspiration is performed for culture and sensitivity as well. This technique is divided into two stages; in the first stage, removal of all implanted hardware with debridement and excision of infected and necrotic tissue is performed followed by implanting an antibiotic loaded spacer. After a spacer retention interval which lasts 6 to 8 weeks with antibiotic treatment of 6–8 weeks, a second stage is performed with spacer removal, debridement, obtaining biopsies for culture and sensitivity, and re-implantation is performed after proper wound healing and low serum C-reactive protein (CRP) and erythrocyte sedimentation rate (ESR) thresholds. The antibiotic therapy is continued after the 2nd stage for another 4–6 weeks. In most cases, a minimum of a 12-week period of systemic antibiotic therapy is recommended [16,17,18,19].

### 2.2. Statistical Analysis

Statistical analysis and charting were performed using the GraphPad Prism software package (Prism 6 for Mac OS X, version 5.0d; La Jolla, CA, USA) and Microsoft Excel for Mac 2011 (version 14.7.7). The distribution of categorical data was described using absolute and relative frequencies. To compare the location of the distribution of a metric variable of two independent groups, the Shapiro–Wilk test was first used to check whether the data of both groups were normally distributed. If the normal distribution assumption was not rejected (*p*-value ≥ 0.1), the comparison was carried out with the *t*-test. In the case of rejection of the normal distribution assumption, the Mann–Whitney-U test was applied. To compare the frequency distributions of a categorical variable of independent groups, the Chi-square test or Fisher’s exact test (if there were expected cell frequencies smaller than five) was used.

## 3. Results

After an average follow-up of 54.5 months (SD = 5.5), a total of 35 patients with an average age of 67.7 years (SD = 8.9) were included in the study. The male gender predominated with 60%. The mean body mass index (BMI) of the total cohort was 28.8 kg/m^2^ (SD = 5.5). The BMI was significantly higher in the one-stage revision group with an average BMI of 30.7 kg/m^2^ (SD = 6.0) than in the two-stage revision group with an average BMI of 26.2 kg/m^2^ (SD = 4.0) (*p* = 0.040). Both groups were preoperatively matched for age and previous diseases. Therefore, no significant differences were found for age (*p* = 0.781) and pre-existing conditions, measured by the Charlson Comorbidity Index (CCI) (*p* = 0.445) (Table 1).

The mean inpatient stay was 20 days (15–17 days, SD 3.5) in the one-stage revision group. The inpatient stays after the two-stage revision group were on average 21 days (16–31 days; SD = 4.1) (*p* = 0.540) after the 1st stay and 18.7 days (14–28 days, SD = 3.6) (*p* = 0.083) after the 2nd stay. When comparing total length of stay including both stages versus one-stage, there was a significant difference, with two-stage revision resulting in a longer length of stay (*p* < 0.05). There were no significant differences for the isolated number of germs in both groups (Table 2).

A total of four complications occurred in both groups. In the one-stage revision group, there were 2 cases of deep vein thrombosis and 2 cases of pulmonary embolism. No further surgery was required in the one-stage revision group. In the two-stage revision group, the following complications occurred: acute kidney failure, deep vein thrombosis, pulmonary embolism, and reoperation due to reinfection. There were no significant differences in mortality between the two groups.

For every patient, the identified germs are shown in Table 3. *Staphylococcus epidermidis* was isolated in both groups most often (one-stage group: 5 times; two-stage group: 2 times). Furthermore, various *Staphylococcus* and *Streptococcus* species were isolated.

The mean Oxford Knee Score in the one-stage revision group was 27 (6–45; SD = 10.7) and in the two-stage revision group was 27.7 (18–42; SD = 7.2) (*p* = 0.928). With regard to pain in the area of the operated knee joint, only 2 patients (5.7%) in the dataset had no pain, with 16 patients (45.7%) having moderate to severe pain. Nine patients of these patients were (47.4%) from the one-stage revision group (Table 4).

The mean EQ-5D-5L Index in the one-stage revision group was 0.634 (range: 0.5–1; SD = 0.342) and in the two-stage revision group was 0.671 (range: 0.42–0.91; SD = 0.161) (*p* = 0.647) (Table 3). There was also no significant difference in the mean SSQ-8 Score (*p* = 0.904) (Table 3). In the evaluation of the SF-36 questionnaires, no significant differences were found between the one- and the two-stage revision group (Table 5).

The mean QALYs were higher for the one-stage revision at 3.26 (SD = 1.41) than for the two-stage revision at 2.89 (SDA = 0.71) (Figure 1). The average difference in QALYs between the one- and two-stage revision group was 0.37 with no statistically significant difference (*p* = 0.395).

## 4. Discussion

Periprosthetic joint infection is a serious complication that can be associated with a reduction in quality of life and significant costs. There are various studies in the literature that present the outcomes after one- and two-stage revisions, as presented in the systematic review from Panguad et al., 2019 [10]. However, the main focus of many studies is on clinical outcomes, particularly the eradication rates and revision rates of both procedures. There are few studies comparing the two surgical procedures. The eradication of infection is crucial when treating PJIs, not only for the surgeon but also for the patient. Another important factor is to achieve the best possible quality of life as well as functionality following the surgery. In this context, no robust data are available in the literature examining clinical outcomes and patient’s function or quality of life measured by PROMs.

In terms of clinical outcomes, our study could not identify any significant differences in complication rates between the one- and two- stage revision. Thiesen et al. (2021) showed that patients with a two-stage revision had significantly higher risks for medical complications [20]. The authors found that acute renal failure was significantly more frequent with a two-stage revision, especially in multimorbid patients with a high CCI index [20]. In our cohort, acute renal failure occurred in only one patient (6.3%) after a two-stage revision. Regarding mortality during the inpatient stay, Thiesen et al. (2021) found no significant differences between the one- and two-stage revision, analogous to our data. A study from the Norwegian registry showed a significantly higher mortality for the two-stage revision [21].

In our study, only one patient required reoperation due to reinfection after a two-stage revision. This corresponds to a periprosthetic knee joint infection cure rate of 93.8%, which is comparable to the current literature [10]. In the single-stage revision group, no reoperation was required. Due to the lack of a 100% response rate, these results need a careful interpretation. Zahar et al. (2016) showed a 93% eradication success rate after ten years with the one-stage revision [9].

The primary goal of this study was to prove that the one-stage revision may lead to a better quality of life for patients. However, this hypothesis could not be confirmed. Tirumala et al. (2021) were able to demonstrate better results for the one-stage revision in periprosthetic hip joint infections [11]. In this study, the one-stage revision was significantly better for all dimensions, such as physical function and mental health [11]. In our study, the SF-36 questionnaire was used to examine several dimensions, including physical and mental health. However, no significant differences between one- and two-stage revision for a periprosthetic knee joint infection could be found.

Walter et al. (2021) showed that patients with a PJI have a significantly lower quality of life compared to normative data, even years after successful surgical treatment [22]. Our study showed that a relatively large number of patients had severe to very severe pain in the operated joint at the time of the last follow-up examination. The quality of life measured by PROMs was not ideal in various cases. Nonetheless, many patients would have undergone the same operation again. Of the total study population, 19 patients (54.3%) would have undergone the surgery again, with a preponderance in the one-stage group (one-stage group: 70.6%; two-stage group: 46.7%). The reason for this could be that patients were generally satisfied with the eradication of the infection despite functional limitations and sometimes severe pain. This means that patients may be grateful that they “survived” the serious, sometimes life-threatening, diagnosis of a periprosthetic joint infection without more serious limitations.

The question that now arises is whether the current PROMs are ideally suited for the evaluation of patients with a periprosthetic joint infection. It may be that special questionnaires with a special focus on periprosthetic join infections need to be established for the evaluation of PROMs. A currently ongoing prospective, randomized multicenter study is planned to compare one-stage and two-stage septic knee revisions [23]. This study is currently ongoing in 7 centers in Denmark. Clinical follow-up of patients will be done using PROM questionnaires, which will be evaluated at planned intervals of 6 weeks to 10 years. In addition to the Oxford Knee Score, the evaluation of EQ-5D questionnaires is planned. Based on these data and further prospective studies, it may be possible to see which of the two procedures lead to a better quality of life for patients.

Our study provides new results. Nevertheless, these data have to be assessed with limitations. On the one hand, the choice of a retrospective/prospective study design proved to be a disadvantage, as no PROMs data were available preoperatively. PROMs data were only obtained at the time of the last follow-up. Therefore, a prospective study with both pre- and postoperative check-ups (preoperative; 3, 6, 12 months postoperatively) would deliver more valid data. Another disadvantage is the low response rate. For many patients, an analysis was not possible because the patients could not be included in the final evaluation for various reasons. Moreover, the type of primary prostheses was not investigated as this did not influence the indication of one- or two-stage septic exchange technique.

## 5. Conclusions

The one-stage revision had a significantly higher proportion of patients who would have undergone the surgery again. The QALYs tended to be better in the one-stage group. However, there was no statistically significant difference in the QALYs between the one- and two-stage group.

Due to a lack of data, further studies, especially prospective randomized trials, need to be done. These studies will hopefully show whether the one-stage revision for treating a periprosthetic joint infection will become the gold standard.

## Figures and Tables

**Figure 1 antibiotics-11-01602-f001:**
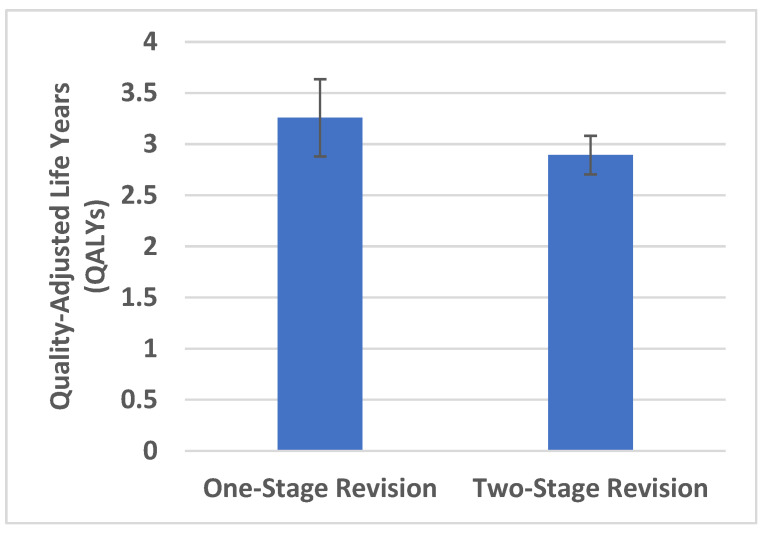
Cumulative quality-adjusted life years (QALYs) for each group. Error bars represent standard error of the mean.

**Table 1 antibiotics-11-01602-t001:** Patient characteristics.

Variables	One-Stage	Two-Stage	Overall	*p*
	(*n* = 19)	(*n* = 16)	(*n* = 35)	
Age (years)	67.7 (SD = 8.5)	67.8 (SD = 9.7)	67.7(SDA = 8.9)	
BMI (kg/m^2^)	30.7 (SD = 6.0)	26.2 (SD = 4.0)	28.8 (SDA = 5.6)	*p* = 0.040
Follow-up (months)	56.5 (SD = 2.7)	51.9 (SD = 7.0)	54.5 (SDA = 5.5)	
Female (%)	31.6 (%)	50 (%)	40 (%)	
Male (%)	68.4 (%)	50 (%)	60 (%)	
CCI	2.1 (SD = 1.6) Min = 0 Max = 5	1.6 (SD = 1.5) Min = 0 Max = 4		*p* = 0.445

**Table 2 antibiotics-11-01602-t002:** Detailed information about the length of stay and germ count between both groups.

Variables	Procedure	Mean	SD	Min	Max	*p*
Inpatient stay (days)	One-stage	20.0	3.5	15.0	27.0	0.540
Inpatient stay after the 1st operation (days)	Two-stage	21.0	4.1	16.0	31.0	
Inpatient stay after the 2nd operation (days)	Two-stage	18.7	3.6	14.0	28.0	0.083
Germ count	One-stage	1.2	0.4	1.0	2.0	0.301
	Two-stage	1.0	0.6	0.0	2.0	

**Table 3 antibiotics-11-01602-t003:** Identified germs (one-stage vs. two-stage).

	One-Stage	Two-Stage
Patient Number	Identified germs	
1	*Streptococcus agalactiae*	*Staphylococcus epidermidis*
2	*Streptococcus oralis*	*Staphylococcus epidermidis*
3	*Escherichia coli*	*Staphylococcus lugdunensis*
4	*Staphylococcus haemoliticus*	*Streptococcus dysgalactiae*
5	*Staphylococcus aureus*	No germ detection
6	*Staphylococcus epidermidis*	*Staphylococcus aureus*
7	*Staphylococcus capitis*	*Streptococcus agalactiae*
8	*Corynebacterium pseudodophteriticum*	*Enterococcus faecalis*
9	*Staphylococcus lugdunensis*	*Micrococcus luteus*
10	*Staphylococcus epidermidis*	No germ detection
11	*Staphylococcus epidermidis*	No germ detection
12	*Streptococcus* (not further defined)	No germ detection
13	*Staphylococcus epidermidis*	*Corynebacterium amycolatum*
14	*Staphylococcus epidermidis*	No germ detection
15	*Staphylococcus* (not further defined)	No germ detection
16	*Streptococcus agalactiae*	No germ detection
17	*Propionibacterium* (not further defined)	
18	*Klebsiella* (not further defined)	
19	*Staphylococcus epidermidis*	

**Table 4 antibiotics-11-01602-t004:** Descriptive statistics: Oxford Knee Score, EQ-5D-5L Index, and SSQ8.

Variables	Procedure	*n*	Mean	SD	Min	Max	*p*
Oxford Knee Score	One-Stage	17	27	10.7	6	45	0.928
	Two-Stage	16	27.7	7.2	18	42	
EQ-5D-5L Index	One-Stage	15	0.634	0.342	0.5	1	0.647
	Two-Stage	15	0.671	0.161	0.42	0.91	
SSQ-8	One-Stage	18	26	5.9	11	34	0.904
	Two-Stage	16	25.8	7.3	15	37	

**Table 5 antibiotics-11-01602-t005:** Detailed information on the results of the SF-36 questionnaire.

Variables	Procedure	*n*	Mean	SD	Min.	Max.	*p*
SF36 Physical functioning	One-Stage	18	0.305	0.236	0.00	0.90	0.055
	Two-Stage	16	0.462	0.238	0.00	0.95	
SF36 Role limitations due to physical health	One-Stage	18	0.347	0.438	0.00	1.00	0.985
	Two-Stage	16	0.328	0.405	0.00	1.00	
SF36 Role limitations due to emotional problems	One-Stage	18	0.815	0.347	0.00	1.00	0.605
	Two-Stage	16	0.688	0.479	0.00	1.00	
SF36_Energy/Fatigue	One	17	0.491	0.190	0.20	0.80	0.330
	Two	16	0.428	0.152	0.20	0.65	
SF36 Emotional well being	One	17	0.649	0.195	0.20	0.88	0.885
	Two	16	0.670	0.134	0.40	0.92	
SF36 Social functioning	One	18	0.576	0.359	0.00	1.00	0.431
	Two	16	0.711	0.208	0.38	1.00	
SF36 Pain	One	18	0.446	0.283	0.10	1.00	0.423
	Two	16	0.516	0.213	0.10	0.88	
SF36 General Health	One	18	0.485	0.212	0.13	0.80	0.691
	Two	16	0.519	0.218	0.20	0.85	

## Data Availability

Data and materials are available, consent to participate was given, consent to publish was given.

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
