# Peer review of "Comparison of Patient-Reported Outcomes Measures and Quality-Adjusted Life Years Following One- and Two-Stage Septic Knee Exchange"

_antibiotics, 2022, doi:10.3390/antibiotics11111602_

Round 1

Reviewer 1 Report

Dear authors,

The manuscript is interesting and provides important information on the patient’s perspective on this debatable topic. I have some comments regarding the article. 

1.      Table 3. is showing the results of different outcome scores between both groups. Please change the title of the Table.

2.      Line 46: Please describe what “selected patients”criteria are considered. Another question is that for selected patients, single-stage was used, while in the other patients which did not meet the “selected patients” criteria you used double-stage?

3.      Please include the types of bacteria found in each of the groups. 

4.      In conclusion, the authors state that the QALYs was better in one-stage, but the results are not statistically significant, therefore such a conclusion is not true.

5.      Where did you describe in results the 70.6% patients who would repeat the single-stage surgery? What is the percentage in the double-stage?

6.      Please discuss the reason of death for the 4 patients.

7.      Use a classification system to stage the PJI in each group (ex. McPherson staging system or similar)

Author Response

Dear Reviewer

Thank you for your comments. Enclosed you will find the corrections marked in green in the text.

  1. Table 3.is showing the results of different outcome scores between both groups. Please change the title of the Table.
    1. Correction can be found in line 179
  2. Line 46: Please describe what “selected patients”criteria are considered. Another question is that for selected patients, single-stage was used, while in the other patients which did not meet the “selected patients” criteria you used double-stage?
    1. Due to its irrelevance, this sentence was deleted
  3. Please include the types of bacteria found in each of the groups. 
    1. A new table with the identified germs was added (Table 3 (line 171)
  4. In conclusion, the authors state that the QALYs was better in one-stage, but the results are not statistically significant, therefore such a conclusion is not true.
    1. Correction can be found in line 264-266
  5. Where did you describe in results the 70.6% patients who would repeat the single-stage surgery? What is the percentage in the double-stage?
    1. Correction can be found in line 236-238
  6. Please discuss the reason of death for the 4 patients.
    1. Correction can be found in line 92-93
  7. Use a classification system to stage the PJI in each group (ex. McPherson staging system or similar)
    1. Correction can be found in line 67

Reviewer 2 Report

The manuscript is very interesting and of great relevance in orthopedic surgery considering the high rate of infections detected which greatly affect the quality of life of patients and the functionality of prosthetic implants. The work is well written and the conclusions in line with the results obtained. The methodological quality is acceptable. However, some additional insight are needed to make it worthy of publication. In particular, it would be interesting to know whether which types of prostheses showed higher failure rates and how this influenced the choice between one- or two- stage exchange; if they consider it appropriate, the author could add , in the introduction, PMID: 33739013 ; DOI: 10.1007/s00264-015-2809-4 .

Author Response

Dear Reviewer

Thank you for your comments. Enclosed you will find the corrections marked in red in the text (line260-261).

Round 2

Reviewer 1 Report

I recommend for publication in current form.